# Retrotransposon Insertion Polymorphisms (RIPs) in Pig Coat Color Candidate Genes

**DOI:** 10.3390/ani12080969

**Published:** 2022-04-08

**Authors:** Zhanyu Du, Enrico D’Alessandro, Yao Zheng, Mengli Wang, Cai Chen, Xiaoyan Wang, Chengyi Song

**Affiliations:** 1College of Animal Science and Technology, Yangzhou University, Yangzhou 225009, China; dx120190120@yzu.edu.cn (Z.D.); mz120180996@yzu.edu.cn (Y.Z.); mx120200833@yzu.edu.cn (M.W.); 007302@yzu.edu.cn (C.C.); wxyan@yzu.edu.cn (X.W.); 2Department of Veterinary Sciences, University of Messina, Via Palatucci, 98168 Messina, Italy; edalessandro@unime.it

**Keywords:** coat color, phenotypic trait, pig breeds, SINE, RIPs, molecular marker

## Abstract

**Simple Summary:**

Coat color is an important phenotypic trait of pig breeds, and a few genes are involved in its regulation. After long-term selection, pig breeds have formed diversified coat color phenotypes (e.g., black, red, white, spotted, brown, belted, two-end black, etc.). To date, several major genes affecting pig coat color phenotypes have been identified. In this view, recent studies suggested a new type of molecular marker known as retrotransposon insertion polymorphisms (RIPs). This marker is considered an ideal molecular marker, showing the potential to be used for population genetic analysis in pigs, with the possibility to be extended to other livestock animals as well. Here, we revealed 42 RIPs in 20 coat color genes in pigs, and some RIPs may be useful in distinguishing different breeds.

**Abstract:**

The diversity of livestock coat color results from human positive selection and represents an indispensable part of breed identity. As an important biodiversity resource, pigs have many special characteristics, including the most visualized feature, coat color, and excellent adaptation, and the coat color represents an important phenotypic characteristic of the pig breed. Exploring the genetic mechanisms of phenotypic characteristics and the melanocortin system is of considerable interest in domestic animals because their energy metabolism and pigmentation have been under strong selection. In this study, 20 genes related to coat color in mammals were selected, and the structural variations (SVs) in these genic regions were identified by sequence alignment across 17 assembled pig genomes, from representing different types of pigs (miniature, lean, and fat type). A total of 167 large structural variations (>50 bp) of coat-color genes, which overlap with retrotransposon insertions (>50 bp), were obtained and designated as putative RIPs. Finally, 42 RIPs were confirmed by PCR detection. Additionally, eleven RIP sites were further evaluated for their genotypic distributions by PCR in more individuals of eleven domesticated breeds representing different coat color groups. Differential distributions of these RIPs were observed across populations, and some RIPs may be associated with breed differences.

## 1. Introduction

Animal pigmentation is one of the most visible and variable traits shaped by natural and/or artificial selection. As a visible phenotypic marker, pigmentation has played an important role in our inheritance understanding, development, and evolutionary theory [1]. In mammals, skin and hair pigmentation is a consequence of cellular activities and cooperation between two cell types, i.e., melanocytes that produce melanins and keratinocytes that distribute melanins throughout the skin [2]. Melanins are the principal pigments of the skin and hair, and their synthesis occurs within membrane-bound organelles of melanocytes called melanosomes [3]. It is commonly known that the synthesis of principal melanins occurs through the melanogenesis process (ko04916). This process is under the complex regulatory control of multiple genes, including *MC1R*, *ASIP*, *MITF*, *DCT*, *TYR*, *TYRP1*, and *KIT*. Besides these genes, the true melanogenesis operating mechanism should be started by forming melanosomes [4]. There are many key molecules participating in intracellular trafficking during melanosome biogenesis. Melanosome structure and function are determined by a cohort of resident transmembrane proteins, many of which are expressed only in pigment cells and localized specifically to melanosomes [5]. All melanosomal proteins are synthesized in the endoplasmic reticulum (ER) and transit through the Golgi complex and the *trans*-Golgi network (TGN) route to melanosomes [6]. Premelanosome protein (PMEL, also known as PMEL17, SILV, or gp100), an integral membrane protein, is targeted from the TGN to the plasma membrane, from where it is endocytosed and delivered to a Stage I melanosome as a melanosome fibroid structural framework [7]. The melanosome-associated G-protein-coupled receptor OA1 is involved as a regulator to assist the formation of PMEL [8]. Although the mature form of a Stage I melanosome is reached at Stage II and Stage III, a balanced environment inside of the Stage I melanosome is needed, which relies on SLC45A2 and OCA2 (also called P protein). These proteins act as ion pumps coupled with V-ATPase exchange cationic concentrations, such as Na^+^, K^+^, Ca^2+^, and H^+^, to regulate internal pH [9,10], while the protein of TYR, TYRP1, and DCT act as cargoes transported by Rab32/Rab38 and Rab27a from the TGN to the melanosome [11], along with the path constructed by the fibrous protein of KIF13A and Spire1/2 [12].

Almost 200 genes associated with color were identified in mice [13]. It has been reported that mutations in these genes may cause pathogenic variants, such as albinism disease in humans [14] (e.g., variable hypopigmentation of the skin, hair, and eyes), ranging from a total lack of melanin to residual or even near-normal melanin production.

In mammals, the similar mutation phenotype with humans also appears in tigers [15], horses [16,17], dogs [18], cattle [19,20], rabbits [21], mink [22], foxes [23], and raccoon dogs [24]. In pigs, the coat color is an important phenotypic characteristic in breeding work. However, one major challenge faced by pig researchers and breeders is merging different breeds’ advantages into one breed. The Sushan pig breeds were developed by Chinese experts using Meishan, Erhualian, and Yorkshire pig, resulted in a white body coat color with significant advantage in growth and reproduction. Similarly, the uniform black coat color with improved meat quality and reproduction observed in the Sutai pigs breed, is a crossbreed between the blood of Duroc and China local breeds (Jiangsu province) [25]. Kijas et al. [26] found that several SNPs in *MC1R* may be a strong candidate marker in different pig breeds. Wu et al. [27] conducted a genome-wide association study and RNA sequencing to demonstrate that *KIT* variants are responsible for diversifying coat color phenotypes, segregating in a Duroc (Landrace & Large White) hybrid pig population.

Retrotransposons are defined as retroelements that replicate by transcription of an RNA intermediate, subsequent reverse transcription, and insertion of a new copy elsewhere in the genome [28]. Retrotransposons, as an important component of plant and animal genomes, can be classified into non-LTR families (including Long Interspersed Nuclear Elements (LINE) and Short Interspersed Nuclear Elements (SINE)) and Long Terminal Repeat elements (LTR, including endogenous retrovirus (ERV)) [29,30]. Retrotransposons occupy one-third to half of the mammal genomes, which are dominated by LINEs and SINEs, followed by LTR retrotransposons [31]. Chen et al. [32] reported that retrotransposons accounted for 37.13% (929.09 MB) of the pig genome, while LINEs, LTRs, and SINEs accounted for 18.52%, 7.56%, and 11.05%, respectively.

Since transposable elements (TEs) were first discovered in maize in the 1940s [33], an increasing number of TEs and retrotransposons have shown more relevant functions in plants and vertebrates’ genomes. However, retrotransposon insertion polymorphisms (RIPs) as DNA markers in a gene are helpful for agricultural production in flax [34], strawberry [35], sweet potato [36], and grape [37]. Rebeca et al. [38] found that detecting RIPs in the intron and exon of particular genes for humans can act as a utility of developing bioinformatic tools to find autism *spectrum* disorder. Abundant polymorphic retrotransposon insertions were already observed in animals such as dogs [39,40] and *Anolis carolinensis* [41]. Three LINE RIP markers were strongly associated with economic traits in 462 Yorkshire pigs [42]. Two RIPs in the *VRTN* (vertebrae development-associated) gene were found in pigs, which showed Hardy-Weinberg equilibrium distributions in most pig populations [43]. Four RIPs in the *GHR* (growth hormone receptor) gene and one RIP in the *IGF1* (insulin-like growth factor 1) gene was identified in pigs. Further analysis revealed that one RIP in the first intron of *GHR* might play a role in the regulation of *GHR* expression by acting as a repressor [44].

In this study, 20 genes related to coat color in mammals were selected. Then, the structural variations (SVs) of these gene sequences derived from retrotransposons were mined by comparative genomic analysis across 17 assembled pig genomes, combined with Repeat Masker annotation. The SVs annotated as retrotransposon insertions were further confirmed by pool-PCR, and their genotypic distributions were detected in eleven pig populations. Therefore, this particular polymorphic molecular marker was explored and intended to provide a theoretical basis for genetic improvement in pigs.

## 2. Materials and Methods

### 2.1. Ethical Statement

The collection of biological samples and experimental procedures involved in this study were approved by the Animal Experiment Ethics Committee of Yangzhou University (No. NSFC2020-dkxy-02, 27 March 2020).

### 2.2. Sequence Acquisition for Coat Color Genes

The detailed information of the genic sequences ASIP, KIT, MITF, PMEL, GPR143, TYR, EDNRB, TYRP1, MLPH, DCT, OCA2, SLC24A5, SLC45A2, Rab27a, Rab32, Rab38, KIF13A, Spire2, Spire1, and MC1R areshown in Appendix A. These genes with their flanking regions (5-kb 5′upstream and 5-kb 3′downstream) from the reference (Duroc) genome (Sscrofa11.1) were downloaded from NCBI Genbank (https://www.ncbi.nlm.nih.gov/genbank/, accessed on 10 September 2021).

Then, 1000 bp upstream and 1000 bp downstream boundary sequences of each genic region were used to Blast the WGS to define the same genomic positions for the 16 assembled nonreference genomes. These genomes represented different types of pigs, including miniature pigs (Tibetan pig, AORO00000000; Wuzhishan, AJKK00000000; Bama, SIDA00000000), lean type pigs (Duroc, Hampshire, LUXS00000000; Berkshire, LUXW00000000; Pietrain, LUXU00000000; Landrace, LUXT00000000; Yorkshire, LUXX00000000), two crossbred pigs (Yorkshire & Landrace & Duroc, NPJO00000000), and fat type pigs (Bamei, LUXV00000000; Jinhua, LUXY00000000; Rongchang, LUXR00000000; Meishan). Two versions of assembled genomes from the same breeds of Meishan (named as Meishan^Beijing^, JABTCQ000000000; Meishan, LUXQ00000000) and Duroc (Duroc^ref^, AEMK00000000 and Duroc^Ninghe^, JACDOW000000000) submitted by different labs were used for SVs identification. The information of the assembled references and nonreference genomes are summarized in Appendix A.

### 2.3. Structural Variation Prediction, Retrotransposon Annotation, and Insertion Polymorphic Prediction

The structural variations were visually inspected based on the multiple alignments of these genes by the ClustalX program (version 2.0, default parameters) [45]. Only the large structural variations (>50 bp) were retained for further analysis. Retrotransposon annotation of these coat color genes and their flanking sequences were performed using RepeatMasker [46] (version-4.1.2, -nolow). Only sequence segments displaying a cutoff score of more than 1000 and longer than 100 bp for the masking repeats were retained for further analysis, and only the structural variations overlapping at least 50 bp with retrotransposons were designated as retrotransposon insertion polymorphic sites (RIPs). The sequence was viewed by the software Jalview (version-2.11.1.5), Bioedit (version-7.0.9.0), and the result of RepeatMasker by the software Sublime Text 4 (version-build 4126).

### 2.4. Animals for RIPs Verification and Genotyping

Six wild pigs and eleven domestic pig breeds (Duroc, Meishan, Erhualian, Fengjing, Landrace, LargeWhite, Sushan, Bama, Wuzhishan, Banna, and Tibetan) were used for the first round of RIP detection by pool-PCR. Three individuals in one pool-PCR and two pools per domestic pig breed and wild breed were applied for RIP detection. Sushan from Jiangsu province is a synthetic breed that includes Meishan, Erhualian, and Yorkshire genetics. Meishan, Fengjing, and Erhualian originating from Jiangsu, Bama from Guangxi, Banan from Guizhou province, Wuzhishan from Hainan province, and Tibetan from Sichuan province are Chinese native pigs. Landrace, Yorkshire, and Duroc pigs are international commercial breeds collected from a breeding farm of Anhui Province. Three wild pig samples were collected from Anhui and Heilong Jiang provinces, respectively. For the second round of RIP PCR detection in population genetic analysis, 11 breeds (Duroc, Meishan, Erhualian, Fengjing, Landrace, LargeWhite, Sushan, Bama, Wuzhishan, Banna, and Tibetan) were used, and 24 individuals were used for each breed. The phenotypic characteristics of the samples we used were highly standardized in the coat color per breeds.

### 2.5. Samples Collection and PCR Analysis

Ear tissue was collected in parallel to agricultural procedures (i.e., pulling in ear tags). Total DNA was isolated from ear tissue with MiniBEST Universal Genomic DNA Extraction Kit by following the manufacturer’s instructions (TaKaRa, Dalian, China). Primer pairs used for RIPs detection were designed based on the flanking sequences of each insertion site in the Sscrofa11.1. We designed primers around 200bp of the insertion or deletion by the software of Oligo (version-7.0). PCR amplifications were carried out in a total volume of 20 µL, containing 50 ng of genomic DNA, 2 × Taq Master Mix buffer (Vazyme, Nanjing, China), and 10 pmol of each primer. PCR amplification was performed with the following cycling conditions: initial denaturation at 95 °C for 5 min, followed by 30 cycles of 95 °C for 30 s, 58 °C for 20 s, and 72 °C for 30 s, and a final extension of 5 min at 72 °C.

Marker: DL2000 (TaKaRa, Dalian, China). PCR products were analyzed by electrophoresis on a 1.5% agarose gel in 1 × TAE buffer. Gels were stained by ethidium bromide and visualized with UV fluorescence. In addition, the PCR amplification products for transposon insertion and deletion alleles of selected RIPs were further verified by sequencing.

### 2.6. PCA and Cluster Analysis of the SINE RIPs

Based on the SINE RIPs identified in this study, the *R* statistics package (version 3.6.3) was used to generate a presence/absence matrix and perform the PCA analysis. On the same dataset, heatmaps and cluster analysis were computed by the use of the *R* package heatmap tool (version 1.0.12) [47], using the “Euclidean” distance method for clustering.

## 3. Results

### Coat-Color Genes’ SVs Revelation and Detection by Pool-PCR in Different Pig Breeds

The sequences of 20 coat-color genes and their flanking regions (5 kb-5′upstream and 5 kb-3′downstream, respectively), which tend to contain most regulatory elements [48], were downloaded or reassembled based on the 16 genome sequences deposited in NCBI as described in the methods section. The genomic coordinates of the 20 analyzed coat-color genes and their flanking sequences are summarized in Appendix A. Structural variations were identified by multiple alignments, using the ClustalX program for each gene. A total of 167 large structural variations (>50 bp) of coat-color genes, which overlapped with retrotransposon insertions (>50 bp), were designated as putative RIPs and obtained; they are summarized in Table 1 and Appendix A.

Then, these predicted RIPs were investigated by the first round pool-PCR using genomic DNA samples from wild pigs (coat color not clear) and eleven domesticated pig breeds, which represented different types of pigs, including definite coat color white breeds: Landrace, LargeWhite and SuShan [49]; black breeds: Fengjing, Erhualian, and Meishan; mini-pig breeds: black (Tibetan and Banna), two-end-black (Bama), and black with white belly and feet (Wuzhishan); red coat color: Duroc. All these domestic breeds are reported in Figure 1. A total of 42 RIPs were confirmed by pool-PCR, with clear polymorphic PCR products across the analyzed DNA samples. The results are shown in Figure 2, and the electrophoresis results and the number format are illustrated in Appendix A and Appendix A. SVs were not encountered in the gene of MC1R and TYRP1, and among these RIPs’ predictions, 10 and 7 confirmed RIPs of Spire1 and KIF13A respectively, which most results detected by pool-PCR.

Relatively on the PCA results (Figure 3), we selected a light coat color breed of Sushan, Landrace, and LargeWhite as Group 1, selected a full dark coat color breed of Meishan, Fengjing, and Erhualian as Group 2, and selected a mini-size but two-end-black of Bama [50], dark coat color breed of Tibetan and Banna as Group 3 (Figure 1). We compared these three groups together to explore 11 RIP-sites’ polymorphism (the predicted RIP sites information are shown in Table 2) between different breeds for further genotyping distinction work carried out via the second round of PCR detecting in 24 individuals per breed.

The expected three genotypes of retrotransposon insertion polymorphisms were named RIP^+/+^, RIP^+/−^, and RIP^−/−^, respectively. GPR143-7 special presents RIP^+/+^ in all 24 individuals of Group 2, but other groups present three gene types in different frequencies. SLC24a5-1 and Rab27a-1 special present RIP^−/−^ in all 24 individuals of Group 3, but other groups present three gene types in different frequencies. Spire1-13 and Rab38-3 special present RIP^−/−^ in all 24 individuals of Group 2 and Group 3 but present RIP^+/−^ in Group 1 (Table 3). We also found one or two breed-particular situations, such as OCA2-11 special presents RIP^−/−^ in Fengjing and Sushan but other breeds present RIP^+/−^; Spire2-1 special presents RIP^−/−^ in Fengjing and Meishan but other breeds present RIP^+/−^; ASIP-12 special presents RIP^−/−^ in Erhualian, Meishan, Bama, Tibetan, and Wuzhishan but others present RIP^+/−^; MITF-6 special presents RIP^−/−^ in Duroc, Erhualian, and Meishan but others present RIP^+/−^ (Table 3).

KIF13A-37 special presents RIP^+/+^ in Fengjing and Bama but presents RIP^−/−^ in others. KIF13A-4 special presents RIP^−/−^ in Duroc and LargeWhite but presents RIP^+/−^ in others. For the gene of KIF13A, we found the RIP site of KIF13A-4 and KIF13A-37 have the same RIP^+/−^ gene type in all 24 individuals of Meishan, Sushan, and Banna, and similarly have a related expression in the genome (Figure 4).

## 4. Discussion

Nowadays, more researchers turn sight to SV exploration work from plant [36] to animal [51] phenotype and even in human healthy [52]. Particularly, RIPs, as important sources of SVs, play important roles in genomic shaping and phenotype variations [53]. In animals, several cases of the coat color phenotype variations derived from RIPs have been reported. Dong et al. [54] reported a case study in Water Buffaloes (*Bubalus bubalis*) in which their genomic analysis results revealed the 165 bp of 50 UTR transcribed from the LINE-1 was spliced into the first coding exon of *ASIP*, resulting in a chimeric transcript. The increased expression of *ASIP* prevents melanocyte maturation, leading to the absence of pigment in white buffalo skin and hairs. Li et al. [55] reported characterization of the endogenous retrovirus insertion in *CYP19A1*, associated with henny feathering in chicken. Finally, David et al. [56] found endogenous retrovirus insertion in the kit oncogene which determined the white and white spotting in domestic cats. Here, RIPs in the twenty coat-color genes (Table 1) were mined by comparative genomic protocol and PCR detection in *Sus scrofa*. Finally, 42 RIPs, which were generated by SINE or ERV insertions, were confirmed by PCR amplification (Table 1). We found that the number of the confirmed RIPs are much less than predicted. Similar result were found in our previous studies [44,57]. We believe that the number of differences between the predicted RIPs and the verified RIPs are due to the assembled levels and quality of genomes, although some genomes were not well-assembled, resulting in false positive RIP prediction. Furthermore, we found most of the confirmed SVs related to SINE insertion or deletion, while ERV only represented one RIP site (Table 2), which is generally agreed with our previous studies, where SINEs are deduced as the major sources of retrotransposon-derived SVs in the pig genome [58], although LINE, LTR, and SINE accounted for 18.52%, 7.56%, and 11.05% of the pig genome, respectively [32].

Based on the pool-PCR detection of RIPs’ result (Appendix A) and the Principal Component Analysis (Figure 3), the RIPs that have at least two polymorphic genotypes contribute more to each PC; for example, pool-PCR results revealed that RIP-ASIP-5, RIP-Spire1-13, RIP-ASIP-12, and RIP-Rab38-3 were not inserted in Bama, Wuzhishan, Tibetan, and Banna, which were separated by PC1, but were inserted in Duroc, Landrace, LargeWhite, and SuShan, which were separated by PC2 from FengJing, Erhualian, and Meishan. The PCA result revealed that these breeds can be generally classified into three distinct clusters, which generally agrees with the known genetic background and histories of these populations. One cluster includes the breeds of Fengjing, Erhualian, and Meishan, which are distributed in the Taihu region (a lake in Jiangsu Province) and share the same origin; thus, they are also called as Taihu pigs. Fengjing and Erhualian are pure grey black pigs, while Meishan pigs display grey black in the main body with four white points (foot ends) [59]. The second cluster represents the lean type breeds, including SuShan, Duroc, Landrace, and LargeWhite. The coat color of introduced breeds, such as Landrace and LargeWhite have whole white body, and Duroc is in red or dark red, while the breed of Sushan is a newly developed Chinese breed that contains a 25% genetic component of Chinese Taihu pigs, 25% Landrace pigs, and 50% LargeWhite pigs, which also has a white coat color [48]. The third cluster contains the Chinese minipig breeds of Bama, Tibetan, Wuzhishan, and Banna, with three distinct coat color patterns: Tibetan and Banna have a pure black coat color, Bama has a white main body with the black head and tail, and the coat color of Wuzhishan dominates by black with a white belly bottom and feet.

To identify the potential relationships between breed differences (such as coat color and body size) and these RIPs, eleven typical polymorphic RIP-sites were selected to further detect their genotypic distributions in at least 24 individuals per breed for 11 pig populations. We found each RIP-sites with at least two polymorphic genotypes among 12 pig breeds, and the SVs related to retrotransposon insertion and deletion were consistent between varieties of breeds after we rechecked our 24 individuals’ detection results per breed compared to the predicted RIP-sites information. It supports that our PCR results and predicted strategy are credible. The RIP-sites detection results, illustrated from Figure 4 and Appendix A, revealed that some SVs in those target coat-color genes may be related to the differences between breeds. Then, we speculated that *Rab27a-1* and *SLC24A5-1* may be related to pig body size due to these two RIP-sites presenting the SVs deletion genotype in Group 1 and Group 2 only but heterozygous genotype in Group 3, which have a smaller body size than others. However, both the RIP-sites of *Rab38-3* and *Spire1-13* may be related to pig breed coat color due to these two RIP-sites presenting the SVs deletion genotype in Group 2 and Group 3 only but heterozygous genotype in Group 1 for which the coat color is lighter than others. *GPR143-7* may be both related to body size and coat color due to this RIP-site presenting the SVs insertion genotype in Group 2 only but heterozygous genotype in Group 1, which has a light coat color, and also a heterozygous genotype in Group 3, which has a smaller body size but dark coat color. In terms of single breed characteristics, we found *Spire2-1* and *KIF13A-4* present the deletion genotype in the pig breeds of FengJing, Meishan, and LargeWhite, respectively, but other breeds present the heterozygous genotype.

Another interesting finding from the present studies indicated that the RIP-sites of *KIF13A-4* and *KIF13A-37* present similar polymorphic genotypes in the breed of Meishan and SuShan, for which we have introduced that the SuShan breed is a hybrid from the Taihu breed, Landrace, and Large White. By the sight of pig evolution, we can speculate that this correlation expression in SuShan may come from Meishan.

## 5. Conclusions

All of the coat-color genes that we chose for this study present an important function in pigmentation. We acquired almost 167 large SVs (>50 bp) target to these genes, which are related to retrotransposon insertion/deletion, and 42 RIPs were really detected in 12 pig breeds. Eleven RIPs that have more than two polymorphic genotypes were chosen to be verified in 24 individuals per 11 breeds to find if these RIPs are special in some breeds.

The findings obtained in this study reveal that all RIPs’ detection results are consistent with our predicted information, and some RIPs, such as *Rab27a-1*, *SLC24A5-1*, *Rab38-3*, *Spire1-13*, and *GPR143-7*, may be good molecular markers to distinguish different breeds on pig breeding and reproduction.

## Figures and Tables

**Figure 1 animals-12-00969-f001:**
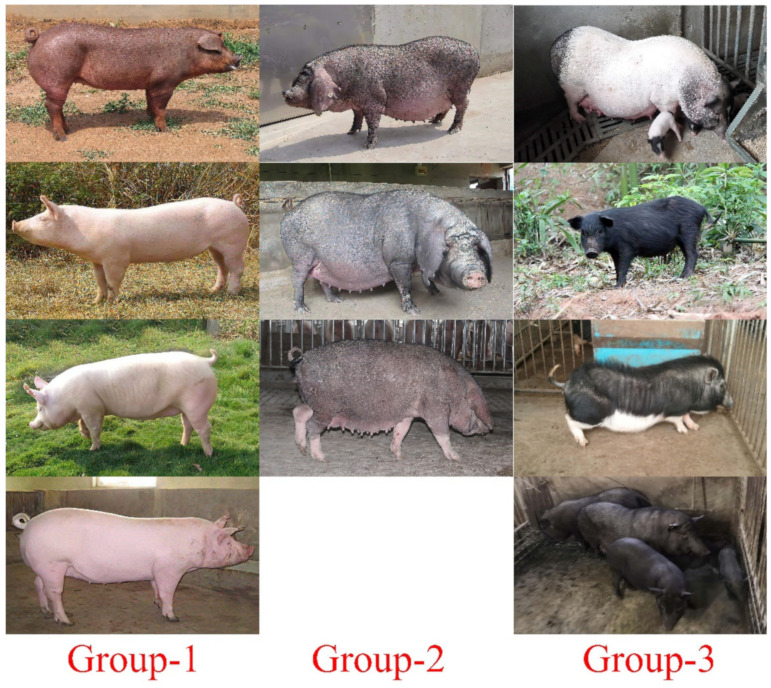
Different coat-color pig breeds. In **Group-1**, from top to bottom is Duroc, Landrace, Large White, and SuShan, respectively. In **Group-2**, from the top down is FengJing, Erhualian, and Meishan, respectively. In **Group-3**, from top to bottom is Bama, Tibetan, Wuzhishan, and Banna, respectively.

**Figure 2 animals-12-00969-f002:**
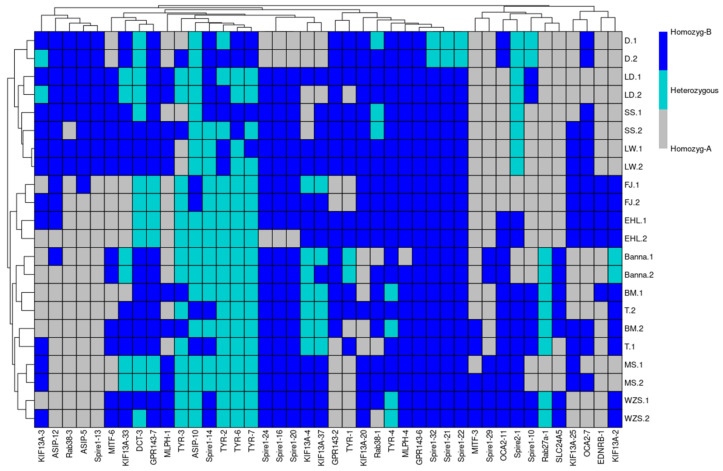
The pool-PCR results shown by heatmap. D (Duroc), LD (Landrace), LW (LargeWhite), SS (SuShan), EHL (Erhualian), FJ (Fengjing), Banna (Banna), BM (Bama), T (Tibetan), MS (Meishan), and WZS (Wuzhishan).

**Figure 3 animals-12-00969-f003:**
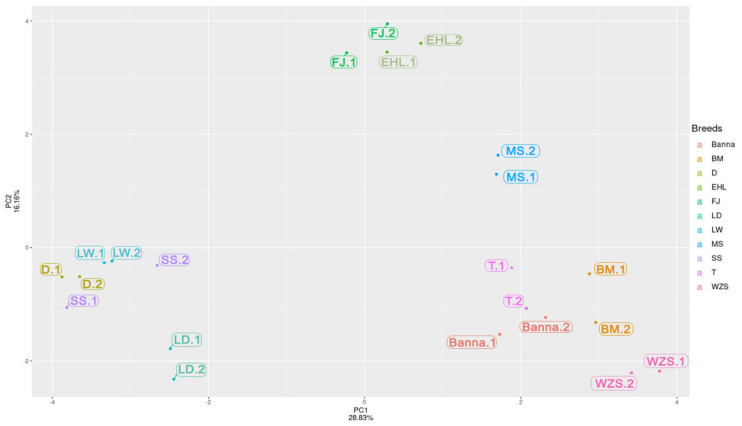
Principal Component Analysis (PCA) based on the results of the pool-PCR result.

**Figure 4 animals-12-00969-f004:**
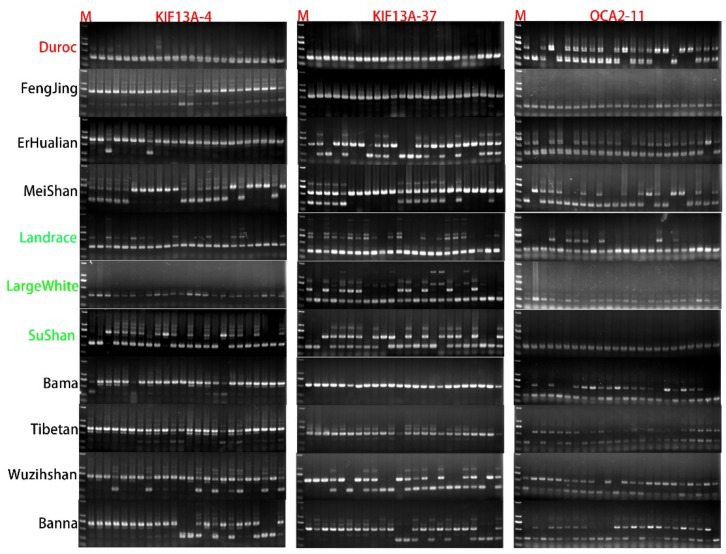
RIP-sites of KIF13A-4, KIF13A-37, and OCA2-11 detection in different small group pig breeds by PCR. M: Marker DL2000.

**Table 1 animals-12-00969-t001:** Predicted large structural variations (SVs) by alignment in coat-color genes and their flanking regions.

Gene Name	Predicted RIPs	Confirmed RIPs
*ASIP*	14	2
*KIT*	8	0
*MITF*	12	2
*PMEL*	1	0
*GPR143*	7	3
*TYR*	7	6
*EDNRB*	2	1
*TYRP1*	0	0
*MLPH*	11	3
*DCT*	3	1
*OCA2*	10	2
*SLC24A5*	1	1
*SLC45A2*	9	0
*Rab27a*	6	1
*Rab32*	2	0
*Rab38*	4	2
*KIF13A*	37	7
*Spire2*	3	1
*Spire1*	30	10
*MC1R*	0	0
**Total Numbers**	167	42

**Table 2 animals-12-00969-t002:** The predicted RIP-sites’ information, which can be detected in different small group pig breeds.

Rip-Sites	Insertion Breeds	Deletion Breeds	Chr	Begin	End	Gene Struture	TE-Type	Length (bp)
ASIP-12	Duroc, Berkshire	The rest of the species	17	37,614,914	37,615,246	Intron-1	SINEA	332
GPR143-7	The rest of the species	Landrace, LargeWhite, Crossbred, Wuzhishan, Bama	X	6,248,783	6,249,085	Intron-9	SINEA	300
MITF-6	Largewhite, Berkshire	The rest of the species	13	51,175,504	51,175,505	Intron-1	SINEB	94
KIF13A-4	The rest of the species	Berkshire, Crossbred, LargeWhite, D-Ninghe, Hampshire, Landrace, Pietrain	7	13,691,836	13,692,190	Intron-1	SINEA	354
KIF13A-37	Bamei, Wuzhishan, Bama, Tibetan, Rongchang	The rest of the species	7	13,497,155	13,497,156	3′flank	ERV III	282
OCA2-11	Wuzhishan, Jinhua, Rongchang, Hampshire	The rest of the species	15	56,806,804	56,806,805	Intron-2	SINEA	339
Rab27a-1	Bama, Tibetan, Bamei	The rest of the species	1	116,538,879	116,538,880	Intron-1	SINEA	326
Rab38-3	LargeWhite, Landrace, Hampshire, Berkshire	The rest of the species	9	21,583,690	21,583,691	Intron-2	SINEA	293
SLC24A5-1	Bama	The rest of the species	1	123,633,504	123,633,505	3′flank	SINEA	310
Spire1-13	The rest of the species	Bamei, MS, Landrace	6	96,966,838	96,967,753	Intron-2	SINEA	370
Spire2-1	Duroc, Tibetan, Largewhite, D-Ninghe, Crossbred, Pietrain	The rest of the species	6	248,403	248,980	5′flank	SINEA	577

**Table 3 animals-12-00969-t003:** The result of eleven RIPs detected in eleven breeds, transformed based on the gel picture.

		Asia-12	GPR143-7	MITF-6	KIF13A-4	KIF13A-37	OCA2-11	Rab27a-1	Rab38	SLC24A5-1	Spire1-13	Spire2-1
Duro	+/+	6	5	0	0	0	6	0	5	0	0	0
	+/−	12	17	0	5	0	14	0	13	0	23	24
	−/−	6	2	24	19	24	4	24	6	24	1	0
Fengjing	+/+	1	24	0	0	24	0	0	0	0	0	0
	+/−	14	0	24	24	0	0	0	0	0	0	0
	−/−	9	0	0	0	0	24	24	24	24	24	24
ErHualian	+/+	0	24	0	22	9	1	0	0	0	0	5
	+/−	0	0	0	2	11	20	0	0	0	0	17
	−/−	24	0	24	0	4	3	24	24	24	24	2
Meishan	+/+	0	24	0	11	11	4	0	0	0	0	0
	+/−	0	0	0	13	13	10	0	0	0	0	0
	−/−	24	0	24	0	0	10	24	24	24	24	24
Landrace	+/+	7	9	7	0	0	0	0	3	0	0	0
	+/−	15	15	10	12	16	7	0	17	0	20	24
	−/−	2	0	7	12	8	17	24	4	24	4	0
LargeWhite	+/+	3	6	16	0	0	0	0	8	0	0	0
	+/−	6	0	8	0	16	24	0	16	0	22	24
	−/−	15	18	0	24	8	0	24	0	24	2	0
SuShan	+/+	4	5	16	3	3	0	0	11	0	0	0
	+/−	11	13	8	12	12	0	0	11	0	14	24
	−/−	9	6	0	9	9	24	24	2	24	7	0
Bama	+/+	0	0	3	19	24	0	13	0	4	0	0
	+/−	0	8	4	5	0	17	11	0	7	0	7
	−/−	24	16	17	0	0	7	0	24	13	24	17
Tibetan	+/+	0	4	0	0	0	0	13	0	0	0	1
	+/−	0	18	22	24	18	23	4	0	7	0	15
	−/−	24	2	2	0	6	1	7	24	14	24	8
Wuzhishan	+/+	0	4	2	15	7	0	18	0	0	0	1
	+/−	0	14	3	9	10	22	3	0	16	0	12
	−/−	24	6	19	0	7	2	3	24	8	24	11
Banna	+/+	0	6	1	16	15	0	18	0	1	0	0
	+/−	4	12	7	4	5	18	5	0	8	0	10
	−/−	20	6	16	4	4	6	1	24	15	24	14

Note: N = 24; “+/+” means Homozygous insert in pig genome; “+/−” means heterozygous insert in pig genome; “−/−” means no insert in pig genome.

## Data Availability

All data needed to evaluate the conclusions in this paper are present either in the main text or the Appendix A.

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
