# Peer review of "Retrotransposon Insertion Polymorphisms (RIPs) in Pig Coat Color Candidate Genes"

_animals, 2022, doi:10.3390/ani12080969_

Round 1

Reviewer 1 Report

The manuscript "Retrotransposon insertion polymorphisms (RIPs) in pig coat color candidate genes" can be an important source of knowledge because pig coat color represents an indispensable part of breed identity. Nevertheless, I have concerns about the number of pigs used in the experiment, because based on electrophoretic gel I see 24 individuals per group, if these animals were related, this number is not representative. The authors should strongly confirm this number of animals is efficient in this study. The manuscript in my opinion is well written and can be published in the Animals but after dissolving these few concerns.
Additional comments/suggestions that should be considered.

Introduction

the authors should add one paragraph, about why the coat color is so important in pigs. This information is shortly given in the abstract, but it should be a little developed in the introduction, please give arguments on the importance of analysis pig coat color.

In the material and method section, as was mentioned before, should be indicated how many pigs were studied in the particular breeds in PCR analysis. Moreover, it is not exactly clear how many individuals were included in the poolPCR analysis. it is written that 3 individuals per pool, which individuals, of which breeds were included pool PCR analysis for domestic pigs? Please dissolve this issue in the material and methods section. it`s can be done on the separate paragraph "animals".

Results

Table 1 I see that you confirmed much less of predicted RIPs, which could be the consequence that you used only three individuals per breed in the pool PCR analysis. However, if your breeds were high standardized in the coat color and you were searching RIP strongly associated with particular coat color, this approach is not bad, but it should be explained. Moreover, because you used much more local breeds (not included in the computative genome searching analysis), they may have additional RIPs in coat color gene-related, which should be also considered.

Figure 4,5 and 6. I think that visualizing all results is not needed. You should be presented with only examples observed in the particular breeds. Moreover, some the electrophoretic gel pictures are low quality and some bands are cut off. These results can be presented on one gel indicating which genotypes were identified in particular breeds or can be presented in the tables with the presentation of band size,  and heterosigosity (n=24) informations can also be given.

Author Response

Review 1

The manuscript "Retrotransposon insertion polymorphisms (RIPs) in pig coat color candidate genes" can be an important source of knowledge because pig coat color represents an indispensable part of breed identity. Nevertheless, I have concerns about the number of pigs used in the experiment, because based on electrophoretic gel I see 24 individuals per group, if these animals were related, this number is not representative. The authors should strongly confirm this number of animals is efficient in this study. The manuscript in my opinion is well written and can be published in the Animals but after dissolving these few concerns.

Additional comments/suggestions that should be considered.

Answer 1: Thanks for the Reviewer’s comment.

The ear samples were collected from sows for each breed from different breeding farms, which are from different families or blood of this breed, and can represent the diversity of this breed. For first round RIP detection, 12 breeds were used, two pools were used for each breed, and each pool includes three individuals, it means that we have six individuals per breed, and these six individuals are from different families of the breed, for the second round PCR detection, 24 individuals were used for each breed, and 11 breeds were used.

We believe that the number differences between predicted RIPs and verified RIPs are due to the assemble levels and quality of genomes, some genomes were not well-assembled, and will generate a lot of false positive RIP predictions, and we have added the explanation in the text. We also tried to detect the RIPs by using more breeds and samples and could not find more RIPs, indicating that the first round RIP detection (12 breeds, two pools and three individuals) and the second round PCR detection by using the number of 24 individuals for 11 breeds are effective for our research.

Introduction

the authors should add one paragraph, about why the coat color is so important in pigs. This information is shortly given in the abstract, but it should be a little developed in the introduction, please give arguments on the importance of analysis pig coat color.

Answer 2: One paragraph to introduce the importance of coat color gene in abstract and introduction, also update the reference order in the final.

In the material and method section, as was mentioned before, should be indicated how many pigs were studied in the particular breeds in PCR analysis. Moreover, it is not exactly clear how many individuals were included in the poolPCR analysis. it is written that 3 individuals per pool, which individuals, of which breeds were included pool PCR analysis for domestic pigs? Please dissolve this issue in the material and methods section. it`s can be done on the separate paragraph "animals".

Answer 3: We thank the reviewer’s comment, and we have reworded in methods.

Sorry for the confusion, for first round RIP detection, 12 breeds were used, two pools were used for each breed, and each pool includes three individuals, which means that we have six individuals per breed, and these six individuals are from different families of the breed.

Results

Table 1 I see that you confirmed much less of predicted RIPs, which could be the consequence that you used only three individuals per breed in the pool PCR analysis. However, if your breeds were high standardized in the coat color and you were searching RIP strongly associated with particular coat color, this approach is not bad, but it should be explained. Moreover, because you used much more local breeds (not included in the computative genome searching analysis), they may have additional RIPs in coat color gene-related, which should be also considered.

Answer 4: Same to answer 1 and answer 3.

We believe that the number differences between predicted RIPs and verified RIPs are due to the assemble levels and quality of genomes, some genomes were not well-assembled, and will generate a lot of false positive RIP predictions, and we have added the explanation in the text.

Figure 4,5 and 6. I think that visualizing all results is not needed. You should be presented with only examples observed in the particular breeds. Moreover, some the electrophoretic gel pictures are low quality and some bands are cut off. These results can be presented on one gel indicating which genotypes were identified in particular breeds or can be presented in the tables with the presentation of band size, and heterosigosity (n=24) informations can also be given.

Answer 4: Thanks, we transformed the 24 individuals detected results of original Figure 4-7 to Table3, and retain the original Figure 7 as Figure 4, also combined original Figure 4-6 in Fig.S3.

Reviewer 2 Report

1) I would like to suggest English proofreading.

2) P3L107: Authors denoted that "Reported rarely be relevant RIP-SV of coat color genes in pigs". Please explain about this in more detail, citing appropriate literatures.

3) PCA analysis: Sizes of characters in Figure 3 should be larger. Please discuss about the contribution of each RIPs to the first and second principal components in detail.

Author Response

Review 2

  • I would like to suggest English proofreading.

Answer 5: We have done proofreading through the whole MS.

2) P3L107: Authors denoted that "Reported rarely be relevant RIP-SV of coat color genes in pigs". Please explain about this in more detail, citing appropriate literatures.

Answer 6: Sorry for the confusion, we deleted this sentence.

3) PCA analysis: Sizes of characters in Figure 3 should be larger. Please discuss about the contribution of each RIPs to the first and second principal components in detail.

Answer 7: Figure 3 has already been rebuilt and replaced. We added discussion in P12L310-L31

Round 2

Reviewer 1 Report

Dear Author I see that all my suggestions were addressed, so I think that this manuscript can be published in this form.

Author Response

Very thanks for your agreement for us to publish our manuscript here.

Reviewer 2 Report

>Answer 7: Figure 3 has already been rebuilt and replaced. We added discussion in P12L310-L31.

I would like to know what "RIPs" more contributed to each PC rather than what breeds were more clearly separated by the PCs.

Author Response

Query:

I would like to know what "RIPs" more contributed to each PC rather than what breeds were more clearly separated by the PCs.

Answer:

We have reviewed the Table 3S and changed the words of discussion in P12L310-L314:

the RIPs that have at least two polymorphic genotypes more contribute to each PC, for example pool-PCR results reveal that RIP-ASIP-5, RIP-Spire1-13, RIP-ASIP-12 and RIP-Rab38-3 not insert in Bama, Wuzhishan, Tibetan, and Banna, which are separated by PC1, but insert in Duroc, Landrace, LargeWhite and SuShan, which are separated by PC2 from  from FengJing, Erhualian and Meishan. 

Round 3

Reviewer 2 Report

The manuscript has been improved.